# Ampullary Large-Cell Neuroendocrine Carcinoma, a Diagnostic Challenge of a Rare Aggressive Neoplasm: A Case Report and Literature Review

**DOI:** 10.3390/diagnostics12081797

**Published:** 2022-07-25

**Authors:** Eleni Karlafti, Maria Charalampidou, Georgia Fotiadou, Ioanna Abba Deka, Georgia Raptou, Filippos Kyriakidis, Stavros Panidis, Aristeidis Ioannidis, Adonis A. Protopapas, Smaro Netta, Daniel Paramythiotis

**Affiliations:** 1Emergency Department, University Hospital of Thessaloniki AHEPA, Aristotle University of Thessaloniki, 546 21 Thessaloniki, Greece; 2First Propaedeutic Department of Internal Medicine, University Hospital of Thessaloniki AHEPA, Aristotle University of Thessaloniki, 546 21 Thessaloniki, Greece; adoprot@hotmail.com; 3First Propaedeutic Surgical Department, University Hospital of Thessaloniki AHEPA, Aristotle University of Thessaloniki, 546 21 Thessaloniki, Greece; geofotiadou@gmail.com (G.F.); st.panidis@gmail.com (S.P.); ariioann@yahoo.gr (A.I.); smaronetta@gmail.com (S.N.); danosprx1@hotmail.com (D.P.); 4Pathology Department, University Hospital of Thessaloniki AHEPA, Aristotle University of Thessaloniki, 546 21 Thessaloniki, Greece; ampou_nt@hotmail.com (I.A.D.); graptou@auth.gr (G.R.); 5Theganio Cancer Hospital of Thessaloniki, 546 39 Thessaloniki, Greece; filipposkyr1@hotmail.gr

**Keywords:** neuroendocrine carcinoma, neuroendocrine neoplasm, large cell, ampulla of Vater

## Abstract

Ampullary large-cell neuroendocrine carcinomas (LCNECs) are extremely rare, and available data are limited on case reports. They present with jaundice, non-specific abdominal pain, or weight loss, imitating adenocarcinoma. Their incidence increases due to the improved diagnostic techniques. However, preoperative diagnosis remains challenging. We report the case of a 70-year-old man with a history of metabolic syndrome, cholecystectomy, and right hemicolectomy, presenting with jaundice. Laboratory results showed increased liver biochemistry indicators and elevated CA 19-9. Esophagogastroduodenoscopy revealed an ulcerative tumor on the ampulla of Vater, and the biopsy revealed neuroendocrine carcinoma. Although computed tomography (CT) detected enlarged regional lymph nodes, the positron emission tomography (PET) showed a hyperactive lesion only in this area. Pylorus-preserving pancreatoduodenectomy with R0 resection was performed. Pathologic evaluation of the 3.1 × 1.9 cm tumor revealed an LCNEC with immunohistochemical positivity at Synaptophysin, EMA, CD56, and cytokeratin CK8/18. The Ki-67 index was 45%. Two out of the nine dissected lymph nodes were occupied by the neoplasm. The patient was discharged home free of symptoms, and adjuvant chemotherapy with carboplatin + etoposide was initiated. A comprehensive review of the reported cases showed that the preoperative biopsy result was different from the final diagnosis in few cases, regarding the subtypes. Conventional radiology cannot identify small masses, and other methods, such as endoscopy, magnetic resonance cholangiopancreatography (MRCP), and FDG-PET scan, might aid the diagnosis. Diagnosis is based on histology and immunohistochemical markers of the surgical specimens. The treatment of choice is pancreatoduodenectomy, followed by adjuvant chemotherapy. However, recurrence is frequent, and the prognosis remains poor.

## 1. Introduction

Periampullary neoplasms include pancreatic, duodenal, distal common bile duct (CBD), and ampullary neoplasms. Ampullary carcinomas arise within the ampulla of Vater [1].

Neuroendocrine neoplasms (NENs) of the gastrointestinal tract (GEP-NENs) are a rare and diverse group. Besides common-shared characteristics, they are also characterized by site-specific distinctions regarding the clinical and pathological course and long-term prognosis [2,3]. Arising from the diffuse neuroendocrine system, they are usually less aggressive than their non-endocrine counterparts [4].

Ampullary NENs are even more uncommonly reported, accounting for less than 0.3% of all gastroenteropancreatic neuroendocrine neoplasms (GEP-NENs) [5,6]. Therefore, available data are based on few case series and retrospective reviews, and ampullary neuroendocrine neoplasms are often discussed together due to the anatomic proximity with those arising from the duodenum or the pancreas, which are more often described [7,8]. In addition, their clinical picture imitates adenocarcinoma (ADC) [9], which is also the most common malignancy in that area [10]. Thus, preoperative diagnosis is difficult.

According to the 2019 WHO classification of tumors of the digestive system, NENs were classified into well-differentiated neuroendocrine tumors (NETs), poorly differentiated neuroendocrine carcinomas (NECs), and mixed neuroendocrine–non-neuroendocrine neoplasms (MiNENs). There have been only few case reports of MiNEN in the ampulla of Vater [11,12,13,14]. Well-differentiated and poorly differentiated NENs reflect two genetically and biologically different entities [15].

NETs were further classified into low-, intermediate-, and high-grade (G1–G2) based on the mitotic rate and the Ki-67 index. Ki-67 labeling reflects high proliferative activity, which has been shown to have a prognostic value. Therefore, G1 NETs are characterized by a low mitotic rate (<2 mitoses/2 mm^2^) and Ki-67 index < 3%, G2 NETs have shown a medium mitotic rate and Ki-67 index, while G3 NETs have a mitotic rate over 20 mitoses/2 mm^2^ and Ki-67 index > 20%. NECs have, by definition, a high mitotic rate (>20 mitoses/2 mm^2^) and Ki-67 index > 20% and are classified as large-cell neuroendocrine carcinoma (LCNEC) and small-cell neuroendocrine carcinoma (SCNEC) [16].

Periampullary neuroendocrine carcinoma (NEC) usually presents with jaundice, bleeding, non-specific abdominal pain, and less commonly with weight loss. Presentation with acute pancreatitis is rare [9].

We provide herein a case report of a large-cell neuroendocrine carcinoma (LCNEC) of the ampulla of Vater, as well as a thorough literature review to consolidate our knowledge. However, the most effective treatment strategies and their exact prognosis remain unclear due to these tumors’ low incidence.

## 2. Case Report

A 70-year-old man presented to the Emergency Department of our hospital because of jaundice, during the last month. His medical history includes colon polyps, right hemicolectomy (two years before he was diagnosed with an adenoma with low-grade dysplasia in caecum), cholecystectomy, inguinal hernia repair, type 2 diabetes mellitus, hypertension, and dyslipidemia. Vital signs on presentation were within normal limits. Abdominal examination revealed reduced bowel movements.

On his admission, the blood biochemistry tests showed a total bilirubin concentration of 14 mg/dL (normal values 0.1–1.2) and elevated levels of liver enzymes (aspartate aminotransferase (AST), 51 IU/L (0–32); alanine aminotransferase (ALT), 78 IU/L (0–32); γ-glutamyltranspeptidase, 628 IU/L (10–60); alkaline phosphatase (ALP), 457 U/L (40–129)). Chest and abdominal radiography were unremarkable. Abdominal ultrasonography disclosed a dilated biliary tree.

An esophagogastroduodenoscopy (EGD) was scheduled for further assessment and revealed a sizable ulcerative polypoid tumor in the ampulla of Vater (Figure 1). To minimize the patient’s symptoms, endoscopic retrograde cholangiopancreatography was performed with plastic stent placement in both the major pancreatic and the bile duct. The patient’s jaundice and liver biochemistry tests were then improved (total bilirubin, 1.7 mg/dL).

Further laboratory tests showed the following results: carbohydrate antigen 19-9 (CA19-9), 259 IU/mL; carcinoembryonic antigen (CEA), 3.54 ng/mL; cancer antigen 125 (CA 125), 9.79 IU/mL; alpha-fetoprotein (aFP), 1.6 ng/mL; prostatic-specific antigen (PSA), 2.17 ng/mL. The blood CEA, CA 125, aFP, and PSA levels were within the normal limits.

An abdominal contrast-enhanced computed tomography scan (CECT) showed cholangiectasis, a dilatated common bile duct (1.4 cm), a dilated major and accessory pancreatic duct, and an indistinct and thickened duodenum wall between the ampulla of Vater and the heterogeneous appearing head of the pancreas. Enlarged lymph nodes with necrosis were found in porta hepatis (1.9 cm) and behind the third part of the duodenum (1.3 cm). A PET scan showed a hyperactive lesion in the ampulla with a high standardized uptake value (SUV max 10.6) (Figure 2).

The surgical resection of the tumor was decided. In the context of preoperative control, a colonoscopy was performed to check for colon polyps and due to the past medical history of caecum tumor. A perianal abscess was incidentally found and treated with surgical incision and drainage.

The patient underwent pylorus-preserving pancreaticoduodenectomy (Traverso-Longmire/modified Whipple) with pancreatojejunostomy and dissection of the lymph nodes.

Macroscopic examination revealed a tumor measuring 3.1 × 1.9 cm located at the ampulla of Vater. A total of nine lymph nodes were dissected.

Regarding microscopic examination, sections of the tumor showed large, mostly uniform, oval tumor cells with irregular nuclei, finely granular chromatin, and indistinct nucleoli. The neoplastic cells showed little pleomorphism (Figure 3a). They were composed of organoid nests or trabeculi, separated by thin fibrovascularseptae. In a few nests, tumor cells showed peripheral palisading, often with a minor component of pseudoglandular formation (Figure 3b). The majority of those formations contained small foci of necrosis. The surface of the neoplasm was ulcerative (Figure 3c). The neoplastic cells were infiltrating the pancreatic parenchyma (Figure 3d). Two out of the nine dissected lymph nodes were occupied by the neoplasm. Surgical margins were clear of tumor cells. The neoplasm was characterized by the expression of synaptophysin, CD56, CK8/18, and EMA (Figure 4). With the marker of cell proliferation Ki67/MIB1, the percentage of positive nuclei was up to 45% (Figure 5).

The final diagnosis was large-cell neuroendocrine carcinoma of the ampulla of Vater. Based on the TNM classification, the patient’s stage was T3 N1 M0 (pStage IIIA). An uneventful postoperative course was observed, and the patient was discharged free of symptoms on day 11 after the surgery. A course of chemotherapy with carboplatin 200 mg + etoposide 100 mg was initiated to treat NEC. Surveillance with physical examination and chest/abdominal/pelvic CT is recommended every 12 weeks for the first year and then every 6 months [17].

## 3. Discussion

GEP-NENs are rare and are mainly well-differentiated. Poorly differentiated carcinomas (SCNEC and LCNEC) are uncommon [18]. The incidence of GEP-NECs was 4.6 cases per 1 million in 2012 [19]. The large-cell subtype constitutes about 60% of GEP-NEC cases, thus being more common than the small-cell subtype [20].

The incidence of NENs increases, which could be due to the better quality of endoscopic and radiological diagnostic techniques, and those tumors can be detected at an earlier stage before they cause specific symptoms [3,21,22,23].

Recent literature emphasizes more in the histopathological subtypes than the TNM staging system to predict the tumor’s aggressiveness and decide on a treatment plan [24].

We found 19 cases of NENs reported in the literature in a two-year time period, from May 2020 until April 2022 (Table 1) [9,14,25,26,27,28,29,30,31,32,33,34,35,36].

There were only two cases of SCNECs [25,27] and one case of LCNEC [26] reported. Like our case, at least three NEN cases [14,31] had elevated carbohydrate antigen 19-9 (CA19-9) levels, although none of them was a poorly differentiated NEC. This field was not answered in the case of the LCNEC. Elevated CA19-9 in serum reflects a poor prognosis and aggressiveness in cases of pancreatic NENs [37]. Although it is also correlated with poor prognosis in ampullary adenocarcinomas (ADCs) [38,39], it has not yet been analyzed in patients with ampullary NENs.

In a population-based study of 20,836 patients with GEP-NENs examining sex differences, it was shown that females were more likely to have NET histological subtype (vs. NEC) and better overall survival (OS) than males. It was suggested that this might be the result of biological differences, which is concordant with preclinical studies examining the favorable effect of estrogen and progesterone receptors’ expression in patients with GEP-NENs [40].

Surface changes of NENs during endoscopy, such as depression, erosion, or ulceration, usually indicate aggressive disease [41]. Conventional radiology can detect a highly vascularized mass and dilated main pancreatic duct (MPD) and common bile duct (CBD) [42]. Since abdominal computed tomography (CT) scan cannot usually identify small ampullary tumors, additional methods, such as endoscopic resonance cholangiopancreatography (ERCP) and magnetic resonance cholangiopancreatography (MRCP), might also aid the diagnosis [43]. In addition, the use of endoscopic ultrasound (EUS) might define the locoregional staging of ampullary NENs [44,45]. However, despite the advancements in imaging technology, preoperative diagnosis of NECs is still challenging [24].

Further imaging is necessary to rule out distant metastases [46]. Although GEP-NETs are slow-growing and have low glycolytic activity, FDG-PET scan might aid in revealing distant metastases and recurrent lesions of highly proliferative tumors, such as NECs [4,47]. A prospective study showed that FDG-PET scan had only 58% sensitivity in diagnosing neuroendocrine tumors. However, it has a high prognostic value, thus predicting the more aggressive carcinomas [48]. Furthermore, in contrast to cases of pulmonary NECs, imaging of the brain is only recommended in symptomatic patients with NECs of the gastrointestinal tract since available data have indicated that central nervous system (CNS) metastases are less common [49,50]. The use of somatostatin receptors is not proven; however, positive imaging could indicate a well-differentiated tumor [49,50]. Since NETs of the Vater’s ampulla and their metastases often express somatostatin receptors, somatostatin receptor scan (SRS) can detect them with an 86% sensitivity [8,43]. However, although somatostatin receptors are often expressed in NET G3, SRS cannot detect the usually somatostatin-receptor-negative NECs [46]. In addition, mutations in the P53 tumor-suppressor gene and deletion of the Rb gene are often present in NECs, indicating an extremely malignant tumor, thus making those markers useful for the differential diagnosis between NET G3 and NECs [16].

Diagnosis of NECs is often complicated and is based on biopsy specimens [49]. Immunohistochemical markers for neuroendocrine differentiation, such as synaptophysin and chromogranin A, are essential to set the diagnosis. However, in poorly differentiated NECs, chromogranin A may be absent, while synaptophysin may be focal or even absent in the small-cell NEC subtype [51]. The absence of both synaptophysin and chromogranin A is exceptional, and in this case, the differential diagnosis of other tumors must be considered [15].

Neuron-specific enolase (NSE) and CD56 marker are in many cases positive in GEP-NENs but have low specificity [51]. While periampullary adenocarcinomas commonly express MUC1 and cytokeratin 7, 2 markers, they are unusual among GEP-NENs [52].

Histologically, LCNECs, like well-differentiated NETs, have a trabecular or organoid growth pattern, rosette formations, salt- and pepper-like chromatin, or peripheral palisading, also keeping features of poorly-differentiated NECs with high mitotic activity and large necrotic areas. However, unlike SCNEC, they have enlarged nucleoli and a lower nuclear-to-cytoplasmic ratio [53,54,55].

It has been proposed that NECs either originate from multipotent stem cells, such as non-neuroendocrine adenocarcinomas, or arise from well-differentiated NENs. How-ever, the first possibility is more likely, as well and poorly differentiated NENs have separate components [49].

Ampullary LCNECs in the literature are limited to 24 case reports before ours, as identified by our review to date (Table 2) [11,18,24,26,56,57,58,59,60,61,62,63,64,65,66,67,68]. Only 14 cases were pure LCNECs [24,26,56,57,59,60,61,62,63,66], while the other 11 had other components [11,18,26,59,60,65,66,68]. Many of these patients had early recurrence with metastases, especially in the liver [24,56,58,60,61,62,63,64,65,66,68].

The literature’s rate of lymph node involvement varies from 46% to 88% for NETs of the Vater’s ampulla [7]. In a retrospective cohort study, more patients with ampullary NETs had increased tumor grade and positive lymph nodes than patients with duodenal or pancreatic NETs. Tumor size was more prominent in patients with ampullary NETs than in duodenal NETs and smaller than in pancreatic NETs. In addition, the median overall survival (OS) was worse in patients with ampullary NETs undergoing surgical resection (5-year OS about 62%) compared to duodenal NETs or pancreatic head NETs (5-year OS about 75%). However, SCNECs and MiNENs were excluded from this study [7].

Therefore, the treatment of choice for poorly differentiated NECs is pancreaticoduodenectomy (PD) with R0 resection, accompanied by lymphadenectomy. In contrast, other procedures, such as endoscopic local resection and surgical ampullectomy, are discussed in the presence of severe comorbidities [69]. Two operation techniques are performed: the classic Whipple operation or the Traverso and Longmire pylorus-preserving pancreatoduodenectomy (PPPD), which seems to be as effective as the classic Whipple [70]. It has been suggested that PD should be performed by experienced surgeons to manage postoperative complications better [42].

Poorly differentiated GEP-NECs have an aggressive progression and tend to develop widespread metastases. Especially, the ampulla of Vater is highly vascularized, which also explains the high rate of distant metastases of ampullary tumors [9]. Therefore, adjuvant chemotherapy should follow surgical excision.

Due to histology and biological behavior similarities, GEP-NENs are treated according to treatment approaches for small-cell lung cancer (SCLC) recommendations. Adjuvant chemotherapy is recommended, while neoadjuvant chemotherapy is still not well-described [71]. Platinum-based chemotherapy, like cisplatin/etoposide or cisplatin/carboplatin combination, is recommended as a first-line treatment for patients with poorly differentiated NECs, if the patient has adequate organ function and clinical status [17,72,73,74]. However, platinum-based chemotherapy had a substantially lower response rate in NECs with a Ki-67 index < 55% [49]. The use of temozolomide- (TMZ), irinotecan-, or oxaliplatin-based schedules as second-line treatment is still uncertain [72]. On the other hand, it has been proposed in the literature that adjuvant chemotherapy provides a survival benefit in patients with resected ampullary adenocarcinoma. However, it is still not common practice for these patients [75].

It has also been suggested that neoadjuvant chemotherapy could benefit patients with esophageal NENs. However, available data about other non-pancreatic gastrointestinal NENs are still limited [76].

We reported a patient with a rare ampullary large-cell neuroendocrine carcinoma (LCNEC) that had obstructive jaundice and elevated CA19-9 levels. The tumor was pictured in the PET scan as a hyperactive lesion in the ampulla of Vater, was treated with PPPD in our surgical department, and is now receiving the fourth cycle of adjuvant chemotherapy.

## 4. Conclusions

Overall, pure ampullary neuroendocrine large-cell carcinomas are extremely rare entities. On the other hand, they can also sporadically coexist with non-neuroendocrine neoplasms, more frequently adenocarcinomas. Currently available data suggest Whipple’s procedure and adjuvant chemotherapy based on the treatment of gastroenteropancreatic and non-gastroenteropancreatic LCNECs. Further issues that will need to be addressed in the future are the targeted neoadjuvant chemotherapy and the use of endoscopic surgical resection of eligible neuroendocrine neoplasms.

## Figures and Tables

**Figure 1 diagnostics-12-01797-f001:**
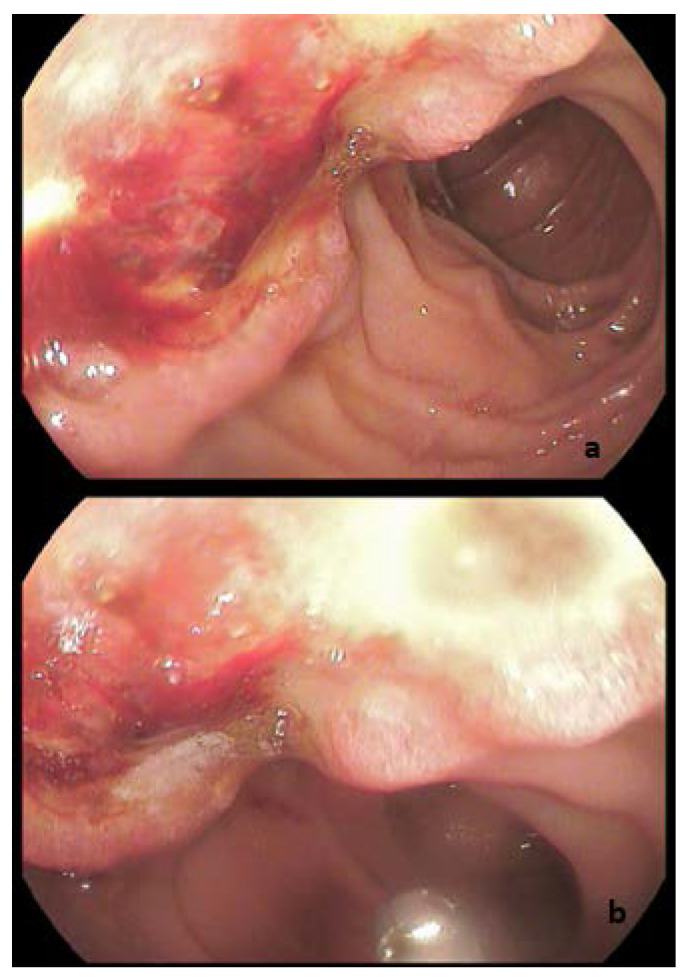
(**a**,**b**) Gastroduodenal endoscopy showed a large ulcerative ampullary lesion.

**Figure 2 diagnostics-12-01797-f002:**
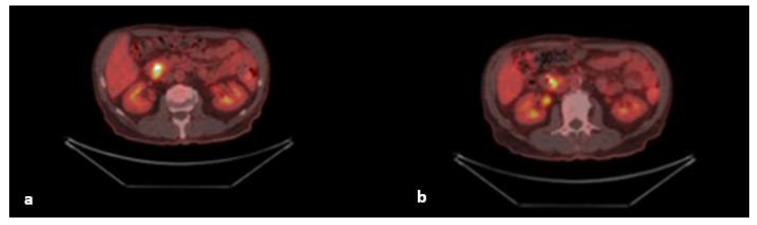
(**a**,**b**) Axial fused PET/CT images show the lesion on the ampullary area. Increased uptake along the common bile duct stent. Focally increased radiopharmaceutical uptake in the intestinal propellers.

**Figure 3 diagnostics-12-01797-f003:**
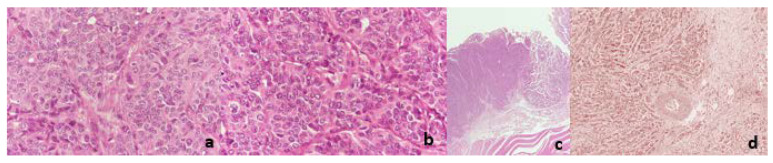
(**a**) (HE, 40×) Large tumor cells with irregular nuclei, finely granular chromatin, and indistinct nucleoli. (**b**) (HE, 40×) Tumor architecture includes organoid nests or trabeculi separated by thin fibrovascularseptae. (**c**) (HE, 2×) The exterior surface of the neoplasm shows ulceration. (**d**) (HE, 40×) Left side of the image shows infiltration of the neoplasm, meanwhile the right side shows involved pancreatic parenchyma.

**Figure 4 diagnostics-12-01797-f004:**
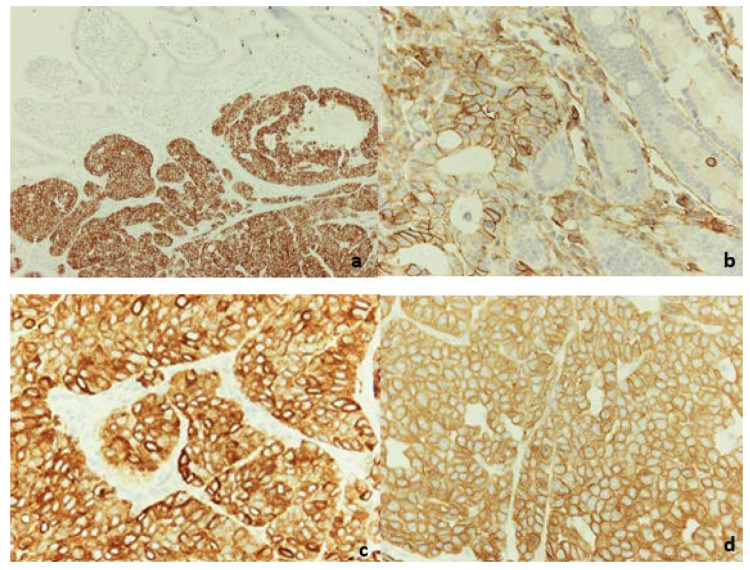
(**a**) Synaptophysin, 10×, (**b**) CD56, 40×, (**c**) EMA, 40×, (**d**) CK8/18, 40×.

**Figure 5 diagnostics-12-01797-f005:**
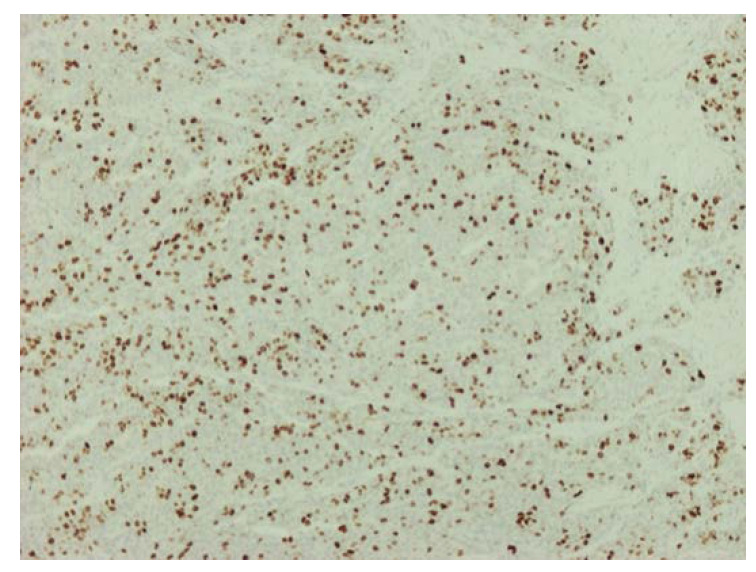
Ki67/MIB1, 10×.

**Table 1 diagnostics-12-01797-t001:** Summary of the literature review of cases of ampullary neuroendocrine neoplasms from May 2020 until April 2022.

	Author	Sex	Age	Clinical Presentation	Tumor Markers in Serum	Distant Metastases on Imaging
1	Kleinschmidt et al. [9]	F	56	right upper quadrant pain	NA	NA
2	Ito et al. [25]	M	65	jaundice	(−)	(−)
3	Jung et al. [26]	M	64	NA	NA	Liver
4	Li et al. [27]	M	57	jaundice, weight loss	(−)	(−)
5	Shiratori et al. [28]	M	60	(−)	(−)	(−)
6	Noorali et al. [29]	F	40	epigastric pain, pruritus	NA	(−)
7	Noorali et al. [29]	M	58	epigastric pain, weight loss	NA	NA
8	Xiao et al. [30]	F	50	pruritus	NA	Umbilical sac
9	Zahid et al. [31]	F	42	jaundice, pruritus	CA19-9 slightly elevated	(−)
10	Guerrero et al. [32]	F	69	epigastric pain	NA	(−)
11	Matli et al. [33]	M	71	abdominal pain, weight loss	NA	NA
12	Matli et al. [33]	F	83	nausea, vomiting	(−)	Liver, left lung, and left shoulder
13	Malhotra et al. [34]	M	33	nausea, vomiting, dyspepsia	NA	(−)
14	Fujii et al. [35]	M	53	presyncope secondary to anemia	(−)	(−)
15	Wang et al. [14]	M	81	jaundice	CA19-9 elevated	(−)
16	Wang et al. [14]	M	69	jaundice	(−)	(−)
17	Wang et al. [14]	M	67	abdominal pain	CA19-9 elevated	(−)
18	Wang et al. [14]	M	72	jaundice, abdominal pain	(−)	(−)
19	Choi et al. [36]	F	46	anemia	(−)	(−)
	**Endoscopic Findings**		**Ki-67 Index**	**Tumor Size**	**Immunohistochemistry**	
1	NA (ERCP/EUS)		3–20%	22 mm	Synaptophysin, Cytokeratin	
2	Unexposed tumor (ERCP)		40–50%	11 mm	Synaptophysin, Chromogranin A, p53, and Rb1	
3	NA		NA	NA	Synaptophysin	
4	NA		40%	50 mm	Synaptophysin, CD56, Cytokeratin	
5	Tumor within the submucosal layer (EUS)		2%	10 mm	Synaptophysin, Chromogranin A	
6	Ulcerative vegetative 22 * 17 mm lesion around the ampulla (EGD)		4%	NA	Synaptophysin, Chromogranin A	
7	Gastritis with a small polyp in the antrum (EGD)		15%	NA	Synaptophysin, Chromogranin A, CEA	
8	21 × 17 mm polypoid ampullary mass (EGD)		NA	11 mm	Synaptophysin, CD56	
9	Ulcerated, tumorous-looking ampulla (ERCP)		NA	NA	Synaptophysin, Chromogranin A	
10	Well-defined, homogeneous, hypoechoic 19 × 9 mm lesion (EUS)		4%	17 mm	NA	
11	Single intramural mass in the area of papilla (EUS)		NA	NA	Synaptophysin	
12	NA (EUS)		NA	NA	Synaptophysin, Chromogranin A, Cytokeratin	
13	Mass lesion causing intra-luminal bulge at the ampulla (EGD)		NA	25 mm	Synaptophysin, chromogranin A, S-100	
14	Bleeding from an erosion		2.50%	11 mm	Synaptophysin	
15	NA		30%	NA	Synaptophysin, CD56	
16	NA (Duodenal endoscopy)		50%	NA	Synaptophysin, Chromogranin A, CD56	
17	Irregular ulcer in the ampulla with a crater-like bulge around		70%	NA	Synaptophysin, Chromogranin A, CD56	
18	NA		70%	NA	Synaptophysin, CD56	
19	Firm, bulging fibrotic ampullary mass with diffuse edema (EGD)		<1%	24 mm	Synaptophysin, Chromogranin A, S-100, NSE	
	**WHO Classification**		**Metastatic Lymph Nodes**	**Adjuvant Chemotherapy**	**Recurrence/Outcome (Months)**	
1	NET G1		1/15	NA	NA	
2	SCNEC		(−)	(irinotecan + cisplatin), (etoposide + cisplatin)	(+)/Death (18)	
3	LCNEC		NA	NA	(+) eyelid metastasis/NA	
4	SCNEC		1	(−)	(+)/Death (14)	
5	NET G1		(−)	NA	(−)/Survival (12)	
6	NET G2		NA	NA	(+) liver metastasis/NA	
7	NET G2		NA	NA	(−)/Survival (84)	
8	NET G1		4/19	NA	NA	
9	NET G3		NA	NA	(−)/Survival (6)	
10	CoGNET		(−)	NA	(−)/Survival (24)	
11	NET		NA	NA	NA	
12	NET G1		NA	(everolimus + lanreotide), temodar, PRRT	NA	
13	NA		NA	NA	NA	
14	NET G1		(−)	(−)	(−)/Survival (89)	
15	MiNEN		NA	(−)	(−)/Death (postoperative complication)	
16	MiNEN		NA	(oxaliplatin + 5-fluorouracil + leucovorin), (etoposide + nedaplatin), (irinotecan + nedaplatin)	(+) liver metastasis/Death (29)	
17	MiNEN		NA	capecitabine + oxaliplatin	(+) liver metastasis/Death (22)	
18	MiNEN		NA	cisplatin + etoposide (hepatic arterial chemoembolization)	(+) liver metastasis/Survival (1)	
19	CoGNET		4/18	(−)	(−)/Survival (36 months)	

NA: not answered; M: male; F: female; CA19-9: carbohydrate antigen 19-9; CEA: carcinoembryonic antigen; EUS: endoscopic ultrasound; EGD: esophagogastroduodenoscopy; ERCP: endoscopic retrograde cholangiopancreatography; NSE: neuron-specific enolase; NET: neuroendocrine tumor; G1: grade 1; G2: grade 2; G3: grade 3; MiNEN: mixed neuroendocrine–non-neuroendocrine carcinomas; CoGNET: composite gangliocytoma/neuroma and neuroendocrine tumor; LCNEC: large-cell neuroendocrine carcinoma; SCNEC: small-cell neuroendocrine carcinoma; PRRT: peptide receptor radionuclide therapy.

**Table 2 diagnostics-12-01797-t002:** Summary of the large-cell neuroendocrine carcinomas reported in the English literature.

	Date of Publication	Author	Sex	Age	Clinical Presentation	Tumor Markers in Serum
1.	2003	Cavazza et al. [56]	F	74	jaundice, anorexia, pruritus	tissue polypeptide antigen, CEA
2.	2004	Hartel et al. [57]	F	44	jaundice, pruritus	(−)
3.	2004	Cheng et al. [58]	F	55	epigastric pain	NA
4.	2005	Nassar et al. [59]	M	61	NA	NA
5.	2005	Nassar et al. [59]	M	75	NA	NA
6.	2005	Nassar et al. [59]	M	84	NA	NA
7.	2005	Nassar et al. [59]	F	50	NA	NA
8.	2005	Nassar et al. [59]	M	77	NA	NA
9.	2005	Nassar et al. [59]	M	80	NA	NA
10.	2005	Nassar et al. [59]	M	55	NA	NA
11.	2005	Nassar et al. [59]	F	68	NA	NA
12.	2006	Huang et al. [60]	M	59	jaundice, anorexia	NA
13.	2006	Selvakumar et al. [61]	M	48	jaundice, anorexia, weight loss	NA
14.	2006	Jun et al. [62]	M	56	anorexia, pruritus	NA
15.	2008	Liu et al. [18]	F	70	jaundice	(−)
16.	2010	Stojsic et al. [63]	M	60	epigastric pain, nausea, vomiting, anorexia, jaundice, fever, and weight loss	NA
17.	2010	Sunose et al. [64]	F	73	jaundice, fatigue	NA
18.	2012	Huang et al. [65]	M	52	jaundice, anorexia, epigastic pain	CEA
19.	2012	Beggs et al. [66]	M	52	obstructive jaundice	NA
20.	2014	Zhang et al. [67]	M	69	jaundice, pruritus	CA19-9
21.	2017	Mahansaria et al. [11]	NA	NA	NA	(−)
22.	2017	Imamura et al. [68]	M	81	(−)	CEA, CA 19-9
23.	2020	Jung et al. [26]	male	64	NA	NA
24.	2021	Sonmez et al. [24]	male	78	jaundice, epigastric pain	NA
	**Distant Metastases on Imaging**	**Endoscopic Findings**	**Biopsy**	**Ki-67 Index**	**Mitotic Rate**	**Maximum Tumor Diameter (mm)**
1.	(−)	Ulceration of the Vater papilla and 3 cm stenosis of the common bile duct	poorly differentiated carcinoma	NA	90/10HPF	NA
2.	(−)	Ulcerated tumor of the main duodenal papilla	NA	NA	4/HPF	16
3.	(−)	Ulcerated polypoidal lesion in the ampullary region	poorly differentiated carcinoma with neuroendocrine features	60%	>50/10HPF	18
4.	NA	NA	NA	NA	NA	NA
5.	NA	NA	NA	NA	NA	NA
6.	NA	NA	NA	NA	NA	NA
7.	NA	NA	NA	NA	NA	NA
8.	NA	NA	NA	NA	NA	NA
9.	NA	NA	NA	NA	NA	NA
10.	NA	NA	NA	NA	NA	NA
11.	NA	NA	NA	NA	NA	NA
12.	(−)	Enlarged ampulla of Vater with an intact mucosal surface	carcinoid tumor	NA	NA	28
13.	(−)	Prominent ampulla	NA	NA	9-11/HPF	20
14.	NA	NA	NA	NA	NA	NA
15.	(−)	Swollen duodenal papilla	poorly differentiated endocrine carcinoma	>90%	NA	10
16.	(−)	15 mm ulcerated tumor in the region of the papilla of Vater	NA	41%	36/10HPF	30
17.	(−)	Enlarged ampulla covered with normal mucosa	(−)	NA	NA	25
18.	NA	Stricture at the ampulla of Vater	NA	NA	>20/10HPF	16
19.	NA	Ulcerated irregular ampulla of Vater	poorly differentiated adenocarcinoma	NA	NA	NA
20.	(−)	NA	LCNEC + ADC	NA	60/10HPF	15
21.	NA	NA	NA	>50%	NA	40
22.	NA	14 mm irregular protruding tumor lesion at the ampulla (duodenal endoscopy)	poorly differentiated adenocarcinoma	89%	32/10HPF	NA
23.	Liver	NA	NA	NA	NA	NA
24.	NA	‘bulging’ tumor appearance invading the orifice of the papilla	NET	80%	NA	15
	**Other Components**	**Immunohistochemistry**	**Number of Metastatic Lymph Nodes**	**Adjuvant Chemotherapy**	**Recurrence/Outcome (Months)**	
1.	(−)	Synaptophysin, Chromogranin A, Cytokeratin, NSE	NA	(−)	(+) liver, L2-L3 vertebral, fluid of peritoneal cavity metastases/died (8)	
2.	(−)	Synaptophysin, Chromogranin A, CEA, Cytokeratins (AE 1 1 3 and cytokeratin 7)	2	NA	NA	
3.	5% vague ADC	Synaptophysin, Chromogranin A, CD117	2/23	(−)	(+) liver, mesenterium, peritoneum metastases/died (6)	
4.	Adenoma	NA	1	NA	(+)/died (15)	
5.	(−)	NA	1	NA	(+)/died (30)	
6.	Adenoma	NA	3	NA	(+)/died (13)	
7.	(−)	NA	5	NA	(+)/died (16)	
8.	Adenoma	NA	1	NA	(−)/alive (17)	
9.	(−)	NA	1	NA	(+)/died (16)	
10.	(−)	NA	4	NA	(−)/alive (10)	
11.	(−)	NA	4	NA	(+)/died (4)	
12.	(−)	Synaptophysin, Chromogranin A, NSE	5/18	cisplatin + cyclophosphamide	(+) liver peritoneum metastases/died (10)	
13.	(−)	Synaptophysin, Chromogranin A, Pancytokeratin	2/13	NA	(+) liver metastases/alive (6)	
14.	NA	NA	NA	(−)	(+) liver metastases, pancreatojejunostomy site/alive (2)	
15.	ADC	Synaptophysin, Chromogranin A, Cytokeratin, CD56	(−)	NA	(−)/alive (1)	
16.	(−)	Synaptophysin, PGP 9.5, NSE, Cytokeratin, CK8, Somatostatin, p27, HDAC1, HDAC2, HDAC3	3/19	etoposide + cisplatin	(+) liver metastases/alive (11)	
17.	ADC, signet-ring cell carcinoma, SCC	Synaptophysin, Chromogranin A, CD56	NA	(−)	(+) liver, bone metastases/died (13)	
18.	pancreatic hepatoid microcarcinoma	Synaptophysin, Chromogranin A, Cytokeratin 7, CAM 5.2	(−)	NA	(+) liver metastases/alive (15)	
19.	(−)	Synaptophysin	1	cisplatin + etoposide + mannitol	(+) liver/alive (20)	
20.	intestinal type adenoma	Synaptophysin	(−)	(−)	(−)/alive (33)	
21.	ADC pancreatobiliary type	Synaptophysin, Chromogranin A, NSE, S-100, MUC 1+, MOC31+	4/39	(−)	(+)/NA	
22.	papillary adenoma	Synaptophysin, Chromogranin A	(−)	(−)	(+) liver metastases/died (11)	
23.	(−)	Synaptophysin	NA	NA	(+) eyelid metastases/NA	
24.	(−)	NA	(−)	(−)	(+) liver metastases/died (8)	

Abbreviations: NA: not answered; M: male; F: female; CA19-9: carbohydrate antigen 19-9; CEA: carcinoembryonic antigen; EGD: esophagogastroduodenoscopy; SCC: squamous cell carcinoma; NSE: neuron-specific enolase; NET: neuroendocrine tumor; ADC: adenocarcinoma; LCNEC: large-cell neuroendocrine carcinoma; HPF: high-power field.

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
