# Peer review of "Ampullary Large-Cell Neuroendocrine Carcinoma, a Diagnostic Challenge of a Rare Aggressive Neoplasm: A Case Report and Literature Review"

_diagnostics, 2022, doi:10.3390/diagnostics12081797_

Round 1

Reviewer 1 Report

1. Has incidence of ampullary NENs really increased that much? These tumors cause symptoms and are much less likely to be found incidentally. In introduction should also mention bleeding as potential presenting symptoms- this is much more common than weight loss, pancreatitis, and nonspecific abdominal pain. 

2. Suggest decreasing length of introduction

3. Was bilirubin elevated at time CA 19-9 of 259 was drawn? Would be unusual for NEC to elevate CA 19-9

4. Was the tumor biopsied at time of endoscopy? Why was PET performed? Is this standard at authors institution? 

5. Lymph node yield of 9 is a little low. 

6. Coloscopy should read colonoscopy

Reviewer 2 Report

Thank you for a well-written manuscript and an interesting case. 

Only one comment. 

Did you perform NGS on tumor tissue? It could be interesting to see if Rb1, MEN1, DAXX and ATRX was expressed normally. 
